# Glucose-Limited Fed-Batch Cultivation Strategy to Mimic Large-Scale Effects in *Escherichia coli* Linked to Accumulation of Non-Canonical Branched-Chain Amino Acids by Combination of Pyruvate Pulses and Dissolved Oxygen Limitation

**DOI:** 10.3390/microorganisms9061110

**Published:** 2021-05-21

**Authors:** Ángel Córcoles García, Peter Hauptmann, Peter Neubauer

**Affiliations:** 1Sanofi-Aventis Deutschland GmbH, 65929 Frankfurt, Germany; angel.corcolesgarcia@sanofi.com (Á.C.G.); peter.hauptmann@sanofi.com (P.H.); 2Chair of Bioprocess Engineering, Department of Biotechnology, Faculty III Process Sciences, Technische Universität Berlin, 10623 Berlin, Germany

**Keywords:** non-canonical branched chain amino acids, scale-down, strain screening, mixed-acid fermentation, pyruvate pulse

## Abstract

Insufficient mixing in large-scale bioreactors provokes gradient zones of substrate, dissolved oxygen (DO), pH, and other parameters. *E. coli* responds to a high glucose, low oxygen feeding zone with the accumulation of mixed acid fermentation products, especially formate, but also with the synthesis of non-canonical amino acids, such as norvaline, norleucine and β-methylnorleucine. These amino acids can be mis-incorporated into recombinant products, which causes a problem for pharmaceutical production whose solution is not trivial. While these effects can also be observed in scale down bioreactor systems, these are challenging to operate. Especially the high-throughput screening of clone libraries is not easy, as fed-batch cultivations would need to be controlled via repeated glucose pulses with simultaneous oxygen limitation, as has been demonstrated in well controlled robotic systems. Here we show that not only glucose pulses in combination with oxygen limitation can provoke the synthesis of these non-canonical branched-chain amino acids (ncBCAA), but also that pyruvate pulses produce the same effect. Therefore, we combined the enzyme-based glucose delivery method Enbase^®^ in a PALL24 mini-bioreactor system and combined repeated pyruvate pulses with simultaneous reduction of the aeration rate. These cultivation conditions produced an increase in the non-canonical branched chain amino acids norvaline and norleucine in both the intracellular soluble protein and inclusion body fractions with mini-proinsulin as an example product, and this effect was verified in a 15 L stirred tank bioreactor (STR). To our opinion this cultivation strategy is easy to apply for the screening of strain libraries under standard laboratory conditions if no complex robotic and well controlled parallel cultivation devices are available.

## 1. Introduction

The scale-up of recombinant protein production processes from laboratory scale to industrial scale reactors often leads to problems, which are basically connected to the longer mixing times in large-scale bioreactors. For instance, for an *E. coli*-based recombinant protein production process a 20% reduction of biomass yield and an increase of by-product formation was reported when scaling up from 3 L to 9 m^3^ [1]. Furthermore, it was reported that scaling up an *E. coli* process from 30 L to 450 L results in biomass reduction and a lower product yield [2]. During fermentation, gradient zones of substrate, dissolved oxygen (DO), pH and other parameters are formed due to inefficient mixing and *E. coli* cells respond to these environmental changes by modulating their metabolism [3]. The standard procedure to grow *E. coli* for recombinant protein production processes in industry is the glucose limited fed-batch operation under aerobic conditions. Although it is recommended to add the glucose feed into a zone of high energy dissipation [4,5], for practical reasons, the concentrated feeding solution is often added from the top of the reactor where the local energy dissipation is low and thus aside from the longer mixing times at large scale, there are local spots or plugs with high glucose content, which are distributed only slowly. Due to the higher metabolic and respiratory activity at higher glucose concentrations, oxygen limitation occurs in the feeding zone at higher cell densities, and the increased glucose consumption in this zone leads to the exhaustion of glucose in other regions of the bioreactor. Due to the high affinity of *E. coli* to glucose—i.e., low K_S_ (Monod substrate affinity constant) value, at higher cell densities there are nearly no zones with intermediate concentrations of glucose, i.e., cells steadily switch feast to famine conditions.

*E. coli* responds to glucose excess and DO limitation by shifting metabolism from oxidative respiration to mixed-acid fermentation [5,6]. Under oscillating conditions, the flux through the glycolysis is stimulated when the cells enter the feeding zone, resulting in high amounts of ATP and NADH as well as in the accumulation of pyruvate. This decreases the need for production of additional energy and reducing equivalents by the Krebs cycle. In this scenario, the NAD^+^ and CoA pools are significantly depleted. However, the glycolytic enzyme glyceraldehyde-3-phosphate dehydrogenase requires NAD^+^ and the pyruvate dehydrogenase complex requires CoA as substrate, respectively. Hence, reoxidation of NADH into NAD^+^ and recycling of CoA must be ensured in order to maintain the glycolytic flux towards pyruvate and acetyl-CoA. Moreover, excess of pyruvate must be consumed. Since the Krebs cycle is inhibited, this is achieved by activating mixed-acid fermentation reactions. While acetate production serves to generate ATP, formation of other mixed-acid fermentation products regenerates NAD^+^ and CoA pools [7]. Furthermore, the intracellular accumulation of pyruvate triggered by transient DO limitation increases the metabolic flux leading to biosynthesis of non-canonical branched-chain amino acids (ncBCAAs) through the sequential keto acid chain elongation from pyruvate to α-ketocaproate over α-ketobutyrate and α-ketovalerate by the actuation of the *leu* operon-encoded enzymes [8]. It was proposed that a strong accumulation of pyruvate is a pre-requisite for α-ketobutyrate formation over this shortcut since α-isopropylmalate synthase shows lower affinity towards pyruvate than the other alternative substrates α-ketoisovalerate and α-ketobutyrate. Similarly, an accumulation of α-ketobutyrate is a pre-requisite for ncBCAA biosynthesis, since the affinity of α-isopropylmalate synthase for α-ketobutyrate is around 20-fold lower compared to its preferred substrate α-ketoisovalerate [9]. NcBCAAs can be secreted to the medium and/or be mis-incorporated into native and recombinant proteins through tRNA mis-aminoacylation during protein translation. This occurs due to the promiscuity of aminoacyl-tRNA synthetases, which might accept the non-canonical amino acids instead of the canonical counterparts due to their chemical similarity (reviewed in [10]). For instance, when norvaline is present, leucyl-tRNA synthetase might react with norvaline instead of the canonical leucine. Such mis-incorporation in recombinant proteins can lead to the production of altered protein variants, having non optimal characteristics.

Scale-down bioreactors have been used in bioprocess development in order to better understand the physiological response of *E.coli* cells to concentration gradients occurring in industrial-scale bioreactors due to inefficient mixing. Numerous scale-down models have been developed, including one-reactor and multi-compartment reactor systems [11]. These scale-down simulators can generate feeding and/or oxygen oscillations in order to mimic scale-up effects. For instance, an accumulation of pyruvate-based amino acids such as the ncBCAAs norleucine and norvaline as well as alanine and valine was reported in a standard stirred tank reactor (STR) fed-batch *E. coli* cultivation under glucose excess and induced DO limitation upon a stirrer downshift [6]. Furthermore, in a recombinant *E. coli* cultivation, the combination of DO limitation and high glucose concentration in the feed compartment of a two-compartment stirred tank-plug flow reactor (STR-PFR) system resulted in a significantly higher norvaline biosynthesis due to pyruvate accumulation [12]. Although these scale-down approaches provide insights into microbial cell responses that are relevant under large-scale conditions, the complexity of the reactor systems and their laborious operation are not suitable for screening experiments. A solution to this was recently presented by Anane et al. [13], in which pulse-based scale-down experiments were performed in parallel mini-bioreactors on a high performance robotic station. The authors were able to demonstrate both the accumulation of norvaline in the cytosol and its incorporation into recombinant mini-proinsulin. However, such a high-end system is very complex to operate and thus not accessible in standard laboratories for screening purposes.

Additionally, the glucose based pulse feeding causes complex effects, connected to (i) the dependence of glucose uptake through the phosphoenolpyruvate -dependent phosphotransferase system (PTS) by the ratio of phosphoenolpyruvate to pyruvate, (ii) different uptake systems which are relevant for the glucose uptake and differently regulated in connection with growth limitation, and (iii) the branching of the glucose flux in the upper glycolytic pathway, e.g., to the pentose phosphate shunt. Therefore, we developed the idea to apply pyruvate pulses. Pyruvate is the core substrate for both mixed-acid fermentation and branched chain amino acid biosynthetic reactions. Both pathways are activated in large-scale reactors. In addition, unlike glucose, pyruvate is transported into the cell through the PrvT and Usp transporters and thus independent from the PTS [14]. Furthermore, its uptake and metabolism is not dependent on the energy status of the cell. In our approach the pyruvate pulses are applied in cultivations which were performed as glucose-limited fed-batch cultures with the enzyme-based glucose delivery strategy (Enbase^®^, [15]). These cultivations provide conditions which are relevant to industrial operation and additionally can be very easily set up for high throughput screenings.

## 2. Materials and Methods

### 2.1. Strain, Plasmid, and Cultivation Medium

*E. coli* K-12 BW25113 containing the plasmid pSW3_*lacI*^+^ (Appendix A) was used in this study. This plasmid expresses a recombinant mini-proinsulin (MPI) under the control of an isopropyl-β-D-thiogalactopyranosid (IPTG)-inducible tac-promoter. The cultivations were performed in mineral salt medium containing (per L): 2.0 g Na_2_SO_4_, 2.468 g (NH_4_)_2_SO_4_, 0.5 g NH_4_Cl, 14.6 g K_2_HPO_4_, 3.6 g NaH_2_PO_4_ × 2 H_2_O and 1.0 g (NH_4_)_2_-H-citrate, 2 mL of a 1.0 M MgSO_4_ solution, and 2 mL of a trace elements solution. The trace elements solution contained (per L): 0.5 g CaCl_2_ × 2 H_2_O, 0.18 g ZnSO_4_ × 7 H_2_O, 0.1 g MnSO_4_ × H_2_O, 16.7 g FeCl_3_ × 6 H_2_O, 0.16 g CuSO_4_ × 5 H_2_O and 0.18 g CoCl_2_ × 6 H_2_O. Additionally, the medium contained the carbon source as described below. Most of the aforementioned chemicals were acquired from Sigma-Aldrich (Munich, Germany), with exception of NH_4_Cl and MnSO_4_ × H_2_O, which were acquired from Merck (Darmstadt, Germany). 

### 2.2. Cultivation Conditions in a Mini-Reactor System

In this study, two different cultivation modes were tested in a Pall Micro24 reactor system (Microreactor Technologies Inc., Mountain View, CA, USA). The first mode is a reference cultivation consisting of a glucose-limited fed-batch cultivation under aerobic conditions. The second mode is the same kind of glucose-limited fed-batch cultivation but additionally pyruvate pulses and transient down-shifts of the oxygen supply were applied in order to trigger DO limitation. 

As pre-culture for the mini-bioreactor cultivations 30 µL of a cryostock containing *E. coli* BW25113 pSW3_*lacI*^+^ were used to inoculate a 250 mL Erlenmeyer flask with 30 mL of 1:3 diluted supplemented mineral salt medium containing 5 g L^−1^ glucose (Merck, Darmstadt, Germany), 0.1 M Na-phosphate buffer (Merck, Darmstadt, Germany) and 100 µg mL^−1^ ampicillin (Sigma-Aldrich, Munich, Germany). The pre-culture was incubated at 37 °C and 220 rpm in the orbital ISF1-X shaker (Adolf Kühner AG, Birsfelden, Switzerland), overnight. OD_600_ at the end of the pre-culture was measured and a given volume was used to inoculate the mini-bioreactors Pall Micro24 with a total starting volume of 5 mL so that initial OD_600_ was 0.4. The mini-reactor medium consisted of a 1:3 diluted supplemented mineral salt medium containing 4 g L^−1^ glucose, 0.1 M Na-phosphate buffer, 100 µg mL^−1^ ampicillin and 1 µL mL^−1^ antifoam Desmophen (Covestro AG, Leverkusen, Germany). For the minibioreactors the 1:3 diluted medium had to be used instead of a standard mineral salt medium, which had been used in the 15 L bioreactors, because we observed salt precipitation which was clogging the oxygen vent. Cultivations were performed at 37 °C and the pH was maintained at 7.0 by automatic control with NH_4_OH (Bernd Kraft GmbH, Duisburg, Germany) and CO_2_. Stirrer speed was set to 800 rpm and DO set-point to 25%, maintaining the last by automatically increasing the oxygen flow into the mini-reactor. After the end of the batch phase (approx. 4 h) 1 mL of a 400 g L^−1^ EnPump 200 solution and 50 µL of a 3000 U L^−1^ glucoamylase solution (Enpresso GmbH, Berlin, Germany) were manually added into each of the mini-reactors, hence starting the fed-batch phase. EnPump 200 is a soluble starch-derived glucose polymer which is hydrolyzed by glucoamylase and thus delivers free glucose molecules over time, ensuring a glucose-limited cultivation mode. In order to generate the 400 g L^−1^ EnPump 200 solution, 25 g EnPump 200 (Enpresso GmbH, Berlin, Germany) powder was dissolved in a 25 mL diluted mineral salt medium, so that components’ concentration in the mini-reactor remained invariable after adding the EnPump 30 min after beginning of the fed-batch phase. The recombinant mini-proinsulin was induced by a single pulse of IPTG (Sigma-Aldrich, Munich, Germany) to a final concentration of 0.5 mM. The fed-batch phase was further maintained for 3.5 h.

For the cultivation with pyruvate pulses and DO limitation, immediately after beginning of the fed-batch phase, a 0.833 g L^−1^ pyruvate (Sigma-Aldrich, Munich, Germany) pulse was manually added into the reactor. During the following 5 min after pyruvate addition, DO set-point was set to 0, so that no oxygen was supplied into the mini-reactor during that period. 30 min after the first pyruvate pulse, expression of recombinant mini-proinsulin was induced by IPTG as described above. After induction, repeated 0.833 g L^−1^ pyruvate pulses were manually performed each 30 min for a total of 5 pulses. Between pulses, DO set-point was re-established to 25%.

### 2.3. Cultivation Conditions in the 15 L Stirred-Tank Reactor

As for the mini-reactor system, two different cultivation modes were tested in a 15 L stirred-tank reactor. 

In order to generate the pre-culture of the reference cultivation 100 µL of a cryostock of *E. coli* K-12 BW25113 pSW3_*lacI*^+^ were used to inoculate an Erlenmeyer flask with 500 mL of supplemented mineral salt medium containing 5 g L^−1^ glucose and 100 µg mL^−1^ ampicillin. The pre-culture was incubated at 37 °C and 220 rpm in an orbital shaker for 12 h. OD_600_ at the end of the pre-culture was measured and a given volume was used to inoculate the bioreactor (Type 880142.8, Nr. 209, Braun Melsungen AG, Melsungen, Germany) with 7 L starting volume reactor so that the initial OD_600_ was 0.4. The reactor medium consisted of mineral salt medium with 5 g L^−1^ glucose, 2 mL antifoam 204 (Sigma-Aldrich, Munich, Germany) and 100 mg L^−1^ ampicillin. Cultivation was carried out at 37 °C and the pH was maintained at 7.0 by automatic control with 25% NH_4_OH. Airflow was set to 7 vvm and DO set-point to 20%, by using a cascade control of the stirrer speed (initial stirrer speed was 800 rpm). Exponential feed was started after the end of the batch phase (approx. 4 h) according to following equation,
(1)$$F\left(t\right)=\frac{q_{s}}{S_{i}}\cdot \left(X_{0}\cdot V_{0}\right)\cdot e^{\mu_{set}\cdot t}$$
where *F(t)* represents the feed rate over time (L h^−1^), *q_s_* the set-point of the specific glucose uptake rate (0.514 g g^−1^ h^−1^), S_i_ the concentration of glucose in the feed solution (442 g L^−1^), *X*_0_ the biomass concentration at the feed start (g L^−1^), *V*_0_ the volume of the reactor at the feed start (L), *µ_set_* the set-point of the specific cell growth rate (0.3 h^−1^) and *t* the time after feed start. The feed solution consisted of mineral salt medium supplemented with 4 mL L^−1^ trace elements solution, 2 mL L^−1^ MgSO_4_ solution (1.0 M), 100 mg L^−1^ ampicillin and 442 g L^−1^ glucose. The exponential feed was continued for 3 h and afterwards expression of recombinant mini-proinsulin was induced by automatic addition of 0.5 mM IPTG over a time period of 30 min. During this time no feed was added into the reactor. After induction, the feed was re-started with a constant flow rate which was equal to the last flow rate achieved in the exponential feeding phase. This constant feed fed-batch phase was maintained for 6 h. A general overview of the cultivation is shown in Appendix A.

The cultivation exposed to pyruvate pulses and DO limitation was performed similar, but after the exponential fed-batch phase a pyruvate solution was added by the feed pump for 5 min into the reactor, providing a final amount of approx. 1 g L^−1^ pyruvate. During this time no glucose feed was added, the airflow rate was temporary set to 0 and the DO cascade control was stopped, so that the DO decreased to zero. Directly after the pyruvate pulse, expression of recombinant mini-proinsulin was induced by automatic addition of 0.5 mM IPTG over 30 min, as described above. During IPTG addition no glucose feed was added and airflow and DO cascade control were re-established to the setpoints before the pyruvate pulse. After induction, successive 1 g L^−1^ pyruvate pulses were applied by 5 min feed intervals every 30 min as described above, for a total of 4 pulses. Between pulses, the constant glucose feeding phase was activated, so that the constant flow rate was equal to the last flow rate achieved in the exponential feeding phase, and airflow and DO cascade control were re-established. Here the constant feed fed-batch phase was continued for 5 h. A general overview of the cultivation is shown in Appendix A.

### 2.4. Cell Growth and Mini-Proinsulin Analysis

Cell growth was monitored by measuring the optical density at a wavelength of 600 nm (OD_600_) in a photometer Ultraspec 2100 pro (Amersham Bioscience, Marlborough, MA, USA). When applicable, samples were diluted with the original medium into an OD_600_ range of 0.3–0.8. Concentration of recombinant mini-proinsulin from hourly samples taken from the cultivations carried out in the STR was analyzed according to an HPLC method internally available at Sanofi-Aventis Deutschland GmbH.

### 2.5. Acetate and Formate Analysis

Acetate and formate concentrations from STR cultivation samples were analyzed offline by enzymatic assays internally available at Sanofi-Aventis Deutschland GmbH.

### 2.6. NcBCAA Analysis

Isolation of the intracellular soluble protein and inclusion body fractions from total cell extracts was carried out as described in the protocol available for BugBuster Protein Extraction Reagent (Merck, Darmstadt, Germany). A certain volume (max. 250 µL) of the isolated protein fractions was mixed up to 1 mL with 5 M HCl (Sigma-Aldrich, Munich, Germany). Resulting solutions were introduced in crystal vials with screw caps resistant to aggressive acids. Closed vials were incubated for 24 h at 80 °C for acid hydrolysis of the cellular proteins. Afterwards, vials were left open in a heating block for 16 to 24 h at 65 °C while rotating until all liquid was evaporated. Hydrolyzed samples were then resuspended with a solution containing 20 mM HCl and 10% isopropanol (Acros Organics ThermoFisher Scientific, Waltham, MA, USA). Resulting solutions were used for further amino acid isolation according to the protocol provided by the EZ:faast^TM^ kit for free (physiological) amino acids by GC-FID (Phenomenex, Aschaffenburg, Germany) with α-aminobutyric acid (Sigma-Aldrich, Munich, Germany) as internal standard. After amino acid isolation, approximately 120 μL of the resulting upper layer were introduced into GC vials and 2 μL were then injected into the GC-FID analyzer 7890A (Agilent Technologies, Santa Clara, CA, USA). Concentration working range was 1–200 μM. The GC was run according to following oven conditions: equilibration time of 0.5 min, 110 °C for 1 min, 30 °C min^−1^ heating up to 320 °C and then 320 °C for 1 min. Each GC run was about 9 min. Nitrogen was used as a carrier gas, with a constant flow rate of 1.5 mL min^−1^. Injection was carried out with a 1:15 split ratio at 250 °C.

NcBCAA concentrations were determined from hourly samples taken from cultivations carried out in the STR. From the cultivations in the mini-reactor system ncBCAA concentrations were just determined at the end of the cultivation (3 h after induction) due to the limited available volumes.

## 3. Results

### Effect of Pyruvate Pulsing and DO Limitation on ncBCAA Biosynthesis and Mis-Incorporation into Recombinant Mini-Proinsulin

It is known that the biosynthesis and mis-incorporation of ncBCAAs into recombinant proteins occurs during *E. coli* based recombinant protein production processes with perturbations in the glucose and dissolved oxygen (see [13]). In this study, we postulate that the repeated application of pyruvate pulses with simultaneous DO limitation in a fed-batch background triggers the synthesis of ncBCAAs in a similar way. Hence, in order to verify our hypothesis, we performed glucose limited fed-batch cultivations with or without pyruvate pulses and analyzed the levels of the ncBCAAs in both intracellular soluble protein and inclusion body fractions of a mini-proinsulin. While the pyruvate pulses with the concomitant DO limitation had no obvious effect on neither the OD_600_ nor on the accumulation of the mini-proinsulin (Figure 1), the concentrations of all three ncBCAAs showed an increase in the intracellular soluble protein fraction after induction of mini-proinsulin production in the cultivations with the pyruvate pulses and concomitant DO limitation (Figure 2). While the concentrations of all of them were similar before induction, norvaline and norleucine showed a stronger increase and higher concentrations than β-methylnorleucine. While the intracellular concentration of all three ncBCAAs increased also in the reference cultivations, the accumulation was much higher in the pulsed cultivations, especially for norvaline and norleucine, while there was only a minor difference in the levels of β-methylnorleucine. Interestingly, while the concentrations of norleucine and β-methylnorleucine increased all the time, norvaline had a maximum level 3 h after induction and dropped then. 

For both tested cultivation types the accumulation of norvaline and norleucine was also observed in the inclusion body fraction (Figure 3), while the concentration of β-methylnorleucine in the inclusion body fraction was below the detection limit. In the inclusion body fraction the accumulation of norleucine was nearly 10-fold higher than that of norvaline, although it was less dependent on the pulses, while the norvaline accumulation in the inclusion body fraction was much higher in the pulsed cultivation compared to the reference. This difference became especially obvious from 3 h after induction onwards, i.e., in the period when the intracellular concentration of norvaline decreased.

As the synthesis of the ncBCAAs is considered as an overflow metabolic reaction from pyruvate, it also was interesting to see the response of mixed-acid fermentation products in these cultivations. For both tested cultivation conditions, the extracellular concentration of acetate was highest at the end of the batch phase (Figure 4a). As expected, it decreased in the first phase of the fed-batch cultivation. A second increase of acetate was observed after induction, while the increase due to the intermittent pyruvate pulses with concomitant DO limitation was not very significant. The difference in acetate concentrations between the pulsed cultivations and the reference becomes only obvious approx. 2 h after induction. The sudden and momentary accumulation of acetate during the fed-batch phase in the reference cultivation corresponds with a sudden glucose accumulation after induction due to the relatively high feeding rate applied (Appendix A).

Unlike acetate, formate is known to accumulate only in response to oxygen limitation. Therefore, as expected, its concentration was low at the end of the batch phase and it disappeared in the first phase of the fed-batch cultivation (Figure 4b). While induction only caused a very slight transient increase of formate, the pyruvate pulses together with the DO limitation caused a very strong increase of formate. As has been previously discussed formate is not easily re-metabolized in standard mineral salt media [16,17] and it thus simply accumulates as an indicator of oxygen limitation. 

As we believe that the use of pyruvate pulses could be valuable for screening of clone libraries, we applied it in a simple mini-bioreactor system, the Pall Micro24 bioreactor. This reactor is a stand-alone 10 mL cultivation system based on 24 bubble columns. While it offers the advantage of performing parallel cultivation, it has no manual continuous feeding option and also sampling is limited. Therefore, in these cultivations we applied the Enbase enzyme-based glucose delivery system, which imitates a glucose limited fed-batch, and we analyzed the inclusion body and intracellular soluble protein fractions at the end of the cultivations. To obtain validated results three biological replicates were performed, i.e., different cultures were performed for each type of cultivation (Figure 5). Pyruvate pulses in combination with DO limitation, which was realized by a sudden decrease of the aeration rate, in these mini-bioreactor experiments also lead to an increase of norvaline by 17.5% (t = 1.14, *p* = 0.32), and norleucine by 51.7% (t = 3.21, *p* = 0.03), while β-methylnorleucine decreased by 14.8% (t = 2.82, *p* = 0.05) in the intracellular soluble protein fraction. In the inclusion body fraction norvaline increased by 25.6% (t = 0.57, *p* = 0.61) and norleucine by 28.1% (t = 1.29, *p* = 0.27). Although the increase is statistically significant only for norleucine in the intracellular protein fraction, the trend for all of these ncBCAAs is similar to the 15 L bioreactor cultivation and also similar to our previous data of the analysis of the free ncBCAAs during production of the same mini-proinsulin where the increase of the level of norleucine was most prominent, while the concentration of norvaline was only about one third of it, and the increase of β-methylnorleucine was lowest [18]. However, it is interesting and deserves further investigations that the level of β-methylnorleucine was much higher in these small-scale cultivations compared to the 15 L bioreactor.

## 4. Discussion

Here we demonstrated that the combination of pyruvate pulses and dissolved oxygen limitation in a fed-batch background can be used as an alternative strategy to simulate the effects of a transient pyruvate accumulation which results in the accumulation of non-canonical branched chain amino acids. This in turn can lead to the mis-incorporation of these amino acids into recombinant proteins and thus to quality problems, especially in pharmaceutical production. We believe that the use of pyruvate compared to glucose bolus addition can have significant advantages as pyruvate is the direct starting point of the metabolic pathway leading to the synthesis of the ncBCAAs. Hence, when adding pyruvate to the cultivation medium, this enters the cell and accumulates intracellularly, rapidly triggering ncBCAA formation through the sequential keto acid chain elongation by the actuation of the enzymes encoded by the *leu* operon. 

Unlike pyruvate, when using glucose pulses, the metabolic pathway is not only longer but also much more regulated. The standard glucose uptake system in *E. coli* is the glucose phosphoenolpyruvate: carbohydrate phosphotransferase system (glucose PTS). In this system the actual glucose uptake rate is directly dependent on the ratio of phosphoenolpyruvate to pyruvate which changes fast during a glucose pulse. This ratio depends not only on the regulation of this uptake system itself and the flux through the upper part of the glycolysis, but also from the flux of pyruvate to acetyl CoA either by pyruvate dehydrogenase or, alternatively under oxygen limitation, over the pyruvate formate lyase aside from other metabolic reactions which include pyruvate. Furthermore, aside from the glucose PTS, glucose is also assimilated by many other transporters, such as the mannose PTS system, the galactose ABC transporter, the galactose permease, and the maltose ABC transporter [19]. 

Also, pyruvate is taken up by different transporters. Kreth et al. [14] discussed the existence of at least three transport systems for pyruvate in order to equilibrate intracellular pyruvate concentrations: an inducible uptake system (Usp system), a constitutive uptake system (PrvT system) and an excretion system. The first two systems were demonstrated to be controlled by catabolite repression. Under cultivation conditions subjected to glucose limitation, pyruvate can be then easily transported into the cell through the PrvT and Usp transport systems. 

Thus, if pyruvate pulses are used, it is not only closer to the ncBCAA route, but also the intracellular pyruvate concentration is regulated by the transporters which should result in more robust system which is less dependent on other environmental conditions if the experiments are performed under glucose limitation. 

In praxis so far, a shift from aerobic conditions to oxygen limitation at either high glucose concentration [6] or in combination with a glucose bolus addition have succeeded in the best production of the ncBCAAs. The latter procedure was applied during investigations which aimed at imitating large scale industrial cultivations in a laboratory environment, i.e., in a scale down simulator [13]. Generally, the time of accumulation of the ncBCAAs was similar in the glucose shift studies and here; it took about one to 1.5 h to see significant changes.

Mis-incorporation of ncBCAAs during translation can lead to the production of altered proteins, having non optimal characteristics. There is plenty of evidence showing that the mis-incorporation of norleucine into recombinant proteins by *E. coli* may lead to alteration of protein structure and biological properties. For instance, recombinant norleucine-substituted β-galactosidase was more resistant to alkylation [20] and recombinant norleucine-rich mammalian calmodulin reported a decreased enzymatic activity in comparison with the canonical variant [21]. When recombinant proteins are to be used as pharmaceutical products for human use, a number of quality criteria have to be fulfilled in order to ensure its effectiveness and safety when delivered in the market. Hence, a number of product parameters have to be tested and these should meet the specifications. One of those parameters is the purity level of the product. The mis-incorporation of ncBCAAs into the product during the recombinant protein production process generates a pool of protein variants differing by only a few amino acids. During downstream processing these variants cannot be easily removed since most protein variants show very similar properties as the main product. Therefore, this issue represents an important concern for the pharmaceutical industry [22]. Also, strategies to suppress mis-incorporation of the no-canonical amino acids by process parameters, such as feeding of leucine or avoiding glucose gradients are not straight forward, especially in industrial large-scale reactors [10]. Hence, the selection of *E. coli* strains showing a better product purity profile is crucial prior to scale-up. However, such screenings have not been extensively applied so far due to methodological limitations. This is where the here proposed method could be applied, which was shown also to function in a simple mini-bioreactor. 

## 5. Conclusions

The cultivation strategy based on the combination of pyruvate pulses and dissolved oxygen limitation in a fed-batch background represents a novel approach to mimic the conditions which lead to the accumulation of ncBCAA and this could be shown in a standard benchtop bioreactor and in a simple parallel mini bioreactor system. We believe the system with pyruvate pulses is so easy that it can be used for the analysis of the ncBCAA potential of whole clone libraries. Such a robustness analysis would be highly important for the development of new large-scale processes of recombinant proteins, to avoid mis-incorporation issues from the beginning. With such a parallel cultivation procedure based on fed-batch conditions in a batch system, bacterial strains showing a better product purity profile, i.e., reduced ncBCAA mis-incorporation, could be selected, thus ensuring a higher product quality.

## Figures and Tables

**Figure 1 microorganisms-09-01110-f001:**
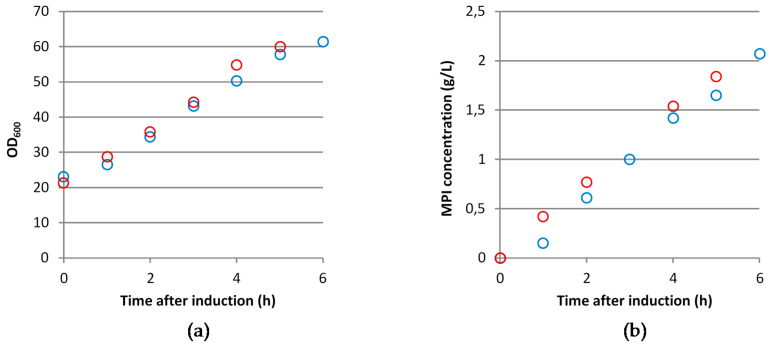
Measured OD_600_ (**a**) and concentrations of recombinant mini-proinsulin (MPI) (**b**) over time after induction of *E. coli* BW25113 pSW3_*lacI*^+^ in a fed-batch cultivation in a 15 L bioreactor under reference cultivation conditions (blue symbols) or with pyruvate pulsing and concomitant dissolved oxygen (DO) limitation (red symbols). Results correspond to the average of three technical replicates. The first measured sample corresponds to the time-point right before induction.

**Figure 2 microorganisms-09-01110-f002:**
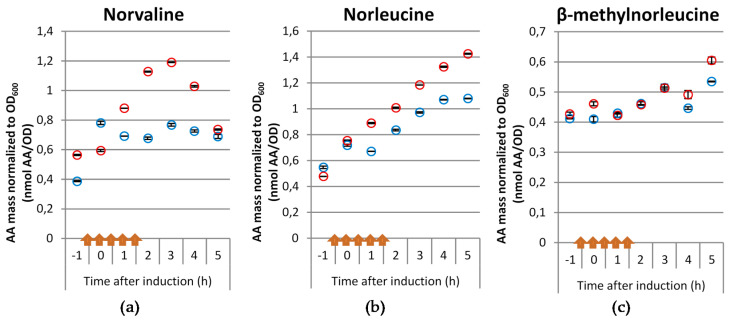
Molar concentrations of norvaline (**a**), norleucine (**b**) and β-methylnorleucine (**c**) normalized to OD_600_ in the intracellular soluble fraction after induction of *E. coli* BW25113 pSW3_*lacI*^+^ glucose limited fed-batch cultivation in a 15 L stirred tank reactor (STR) under reference cultivation conditions (blue symbols) or in a cultivation with pyruvate pulses and concomitant dissolved oxygen (DO) limitation (red symbols). Arrows indicate time points where 1 g L^−1^ pyruvate pulse combined with 5 min DO limitation was applied. Results correspond to the average of 3 technical replicates.

**Figure 3 microorganisms-09-01110-f003:**
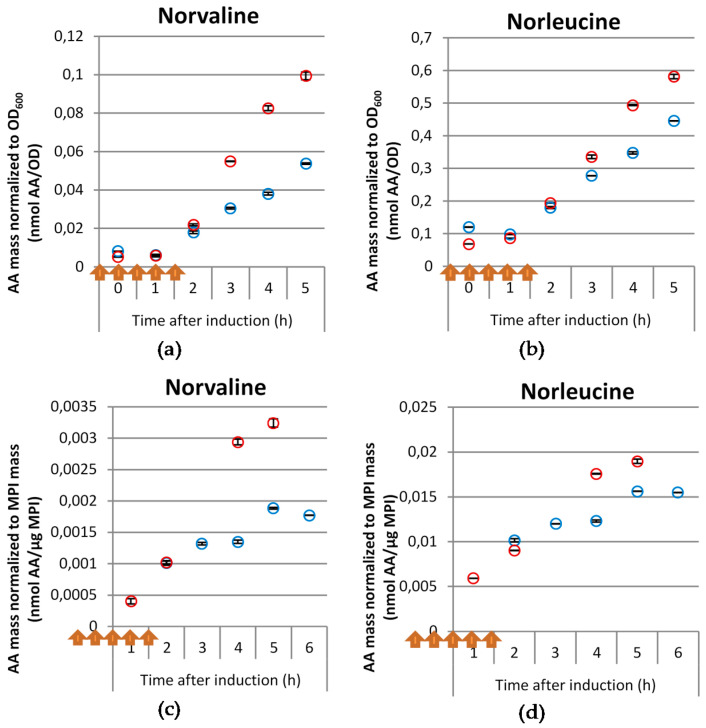
Molar concentrations of norvaline (**a**,**c**) and norleucine (**b**,**d**) normalized to OD_600_ (**a**,**b**) or to recombinant mini-proinsulin (MPI) mass (**c**,**d**) in the inclusion body fraction after induction of *E. coli* BW25113 pSW3_*lacI*^+^ glucose limited fed-batch cultivation in a 15 L stirred tank reactor (STR) under reference cultivation conditions (blue symbols) or in a cultivation with pyruvate pulses and concomitant DO limitation (red symbols). Arrows indicate time points of 1 g L^−1^ pyruvate pulses combined with 5 min DO limitation. Results correspond to the average of 3 technical replicates.

**Figure 4 microorganisms-09-01110-f004:**
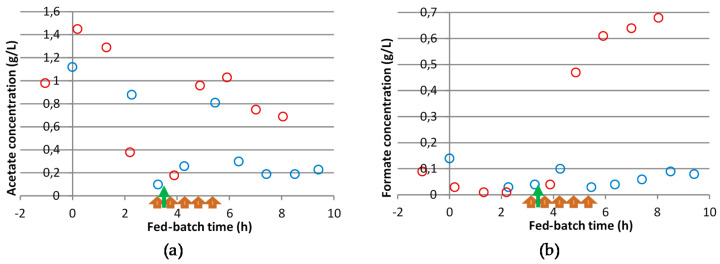
Concentration of acetate (**a**) and formate (**b**) in culture broth supernatants during the feeding phase of a fed-batch cultivation *E. coli* BW25113 pSW3_*lacI*^+^ in a 15 L bioreactor under reference cultivation conditions (blue symbols) or under concomitant pyruvate pulsing and dissolved oxygen (DO) limitation (red symbols), respectively. Orange arrows indicate times of 1 g L^−1^ pyruvate pulses each combined with 5 min DO limitation. Green arrow shows the time of IPTG induction. Results represent the average of 3 technical replicates.

**Figure 5 microorganisms-09-01110-f005:**
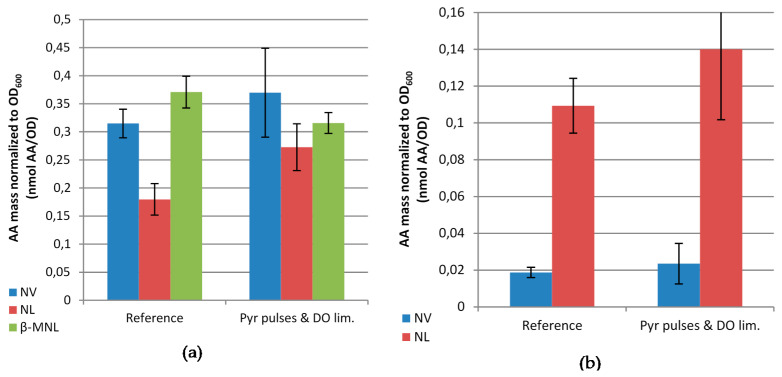
Molar concentrations of norvaline (NV), norleucine (NL) and β-methylnorleucine (β-MNL) normalized to OD_600_ in the hydrolyzed intracellular soluble protein fraction (**a**) and inclusion body fraction (**b**) at 3 h after induction of *E. coli* BW25113 pSW3_*lacI*^+^ cultivation in 10 mL mini-bioreactor cultivations under reference conditions (Reference) or under simultaneous pyruvate pulsing and dissolved oxygen (DO) limitation (Pyr pulses and DO lim.). Results represent the average of 3 independent biological replicates.

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
