# Peer review of "Glucose-Limited Fed-Batch Cultivation Strategy to Mimic Large-Scale Effects in Escherichia coli Linked to Accumulation of Non-Canonical Branched-Chain Amino Acids by Combination of Pyruvate Pulses and Dissolved Oxygen Limitation"

_microorganisms, 2021, doi:10.3390/microorganisms9061110_

Round 1
Reviewer 1 Report
The authors properly answered my comments.
Reviewer 2 Report
Dear Authors
You have answered most of the open points and questions but I still disagree with your arguments on the reference data. Based on already published papers I conclude that you are dedicated experts in scale-up, scale down issues and consequently I assume that you have all the required expertise and equipment at hand to design and run reference experiments in a much better way.
With such experiments you could clearly show if your concept works or not without any need for vague discussion. This is reason why I still recommend rejection of our manuscript
Minor comments:
The error bars in Fig.1 are still missing and the resolution of all Figures is still bad.